# Effect of Intense Exercise on Plasma Macrominerals and Trace Elements in Lidia Bulls

**DOI:** 10.3390/vetsci8060097

**Published:** 2021-06-01

**Authors:** Francisco Escalera-Valente, Marta E. Alonso, Juan M. Lomillos, Vicente R. Gaudioso, Ángel J. Alonso, José Ramiro González-Montaña

**Affiliations:** 1Academic Unit of Veterinary Medicine, Autonomous University of Nayarit, Tepic 69130, Nayarit, Mexico; fescalera@uan.edu.mx; 2Animal Production Department, Veterinary Faculty, University of León, 24071 León, Spain; marta.alonso@unileon.es (M.E.A.); v.gaudioso@unileon.es (V.R.G.); 3Production and Animal Heath, Public Health Veterinary and Science and Technology of Food Department, Veterinary Faculty, Cardenal Herrera—CEU University, 46115 Valencia, Spain; juan.lomillos@uchceu.es; 4Medicine, Surgery and Anatomy Veterinary Department, Veterinary Faculty, Campus de Vegazana, University of León, 24071 León, Spain; ajalod@unileon.es

**Keywords:** exercise, Lidia cattle, mineral metabolism, macrominerals, trace elements

## Abstract

Minerals are inorganic substances present in all body tissues and fluids that directly or indirectly influence the maintenance of multiple metabolic processes and, therefore, are essential for the development of various biological functions. The Lidia bull breed may be considered an athlete, as during a bull fight it displays considerable physical effort of great intensity and short duration in a highly stressful situation. The objective of this study was to assess plasma minerals concentration (macro- and microminerals) in Lidia bulls after intense physical exercise during a bull fight. Plasma Ca, Mg, P, K, Na, Fe, Cr, Co, Ni, Cu, Zn, Se and Mo concentrations were measured in 438 male Lidia bulls. Ca, P and Mg were measured using a Cobas Integra autoanalyzer, while Na and K were determined by ICP-AES, and Fe, Cr, Co, Ni, Cu, Zn, Se and Mo were measured by ICP-MS. All macrominerals, (Ca: 2.96 ± 0.31, Mg: 1.27 ± 0.17, P: 3.78 ± 0.65, K: 7.50 ± 1.58, Na: 150.15 ± 19.59 in mmol/L), and Cr (1.24 ± 0.58), Ni (0.249 ± 1.07), Cu (22.63 ± 4.84) and Zn (24.14 ± 5.59, in μmol/L) showed greater mean values than the reported reference values in the published literature, while Co (0.041 ± 0.07), Se (0.886 ± 0.21) and Mo (0.111 ± 0.08, in μmol/L) values were lower than those reported for other bovine breeds. These increased concentrations could be justified mainly by muscle cell metabolism, hepatic need to provide energy, and intense dehydration and hemoconcentration by losses through sweat glands or urination.

## 1. Introduction

The Lidia bull breed (*Bos taurus brachiceros*) is characterized by its natural aggressiveness and resistance to traditional handling procedures [1]. The Lidia bull’s performance during a bullfight could be compared to that of an athletic animal, due to the intense exercise performed in an unfamiliar and highly stressful environment.

Minerals are essential for multiple metabolic processes; therefore, it is crucial that they remain in balanced concentrations, since a lack or excess can significantly alter many biological functions. Macrominerals are present at larger levels in the animal body or required in larger amounts in the diet, and include calcium (Ca), magnesium (Mg), phosphorus (P), potassium (K), chloride (Cl), sodium (Na) and sulphur (S). However, microminerals, also called trace minerals, meaning they are required in smaller amounts in the animal’s diet, include chromium (Cr), cobalt (Co), copper (Cu), iron (Fe), molybdenum (Mo), nickel (Ni), selenium (Se) and zinc (Zn) [2].

Sodium, chloride, magnesium, potassium, phosphorus, calcium and iron are involved in various basic physiological processes, and their imbalance could alter the animal’s homeostasis [3,4]. Na, Cl, Mg, Ca, K and P are important for the correct conduction of the electrical impulse in the nervous system, as well as in muscular contraction [5,6], while other minerals, including macrominerals, such as iron, selenium, cobalt and copper are fundamental components of enzymes, coenzymes and hormones [6,7].

Both physical exercise and unfamiliar situations may cause significant stress and involve multiple metabolic pathways. Thus, in horses during intense exercise, fluids and electrolytes are lost through sweating [8] and prolonged loss of fluids results in hemoconcentration, hypovolemia and an imbalance in the concentration of different blood parameters. When the deficit worsens, clinical signs of dehydration may appear, with a detrimental effect on physiological processes, lowering sports performance and even compromising the animal’s survival [8,9].

Several studies have been carried out demonstrating that mineral blood parameters are modified in response to intense physical exercise in different species [8,10,11,12,13], for example horses [8,9,10,13], dogs [14,15], donkeys [16], camels [17,18] and bulls [19,20,21,22]. To our knowledge, no studies have measured microminerals in the Lidia breed after intense and stressful exercise.

Therefore, the aim of the present study was to obtain blood mineral values in bulls after intense exercise. Knowledge about mineral changes could improve our understanding of the Lidia bull, and perhaps extend to other animals in the wild that are difficult to handle during stressful situations and, thus, better manage their welfare.

## 2. Materials and Methods

A total of 438 four- to five-year-old male Lidia bulls from different farms who fought in León, Burgos, Valladolid and Salamanca bullrings located in the Autonomous Community of Castilla and León, Spain, were used in the present study. Before the bullfight, the bulls were inspected by the Official Veterinary Services of each bullring to check their suitability for the fight, and to ensure that they were in good health. After the bullfight, all the animals were slaughtered under local regulation law [23].

The bulls were transported from the breeding farms to the bullring (Arena place), always in accordance with the legislation on transport and animal welfare [24] in the days prior to the bullfight. Water and feed are provided there until a few hours before the fight, although this can be maintained if the weather is very hot. The fight of each bull lasts around 15 min.

Exsanguinated blood samples were collected in heparinized vacuum tubes (9 mL) immediately post slaughter. The blood was centrifuged at 4000 rpm (2200× *g*) for 10 min (following the methodology described by Escalera-Valente et al.) [20]. The plasma, free from impurities, was immediately placed in Eppendorf tubes and stored; it was first refrigerated at 4 °C for a maximum of 3 h and then frozen at −20 °C, until it was analyzed in the Instrumental Techniques Laboratory of the University of León.

Ca, Mg, Mg and P were measured using a Cobas Integra 400 autoanalyzer (Roche, Basel, Switzerland) and Roche Diagnostics reagents (Rotkreuz, Switzerland). Na and K were determined using a 1:100 dilution and the samples were analyzed by inductively coupled plasma atomic emission spectrometry (ICP-AES, Optima 2000DV, PerkinElmer Instruments, Waltham, Massachusetts, USA) after the addition of 5 ppm Sc, used as internal standard. The detection limit was 0.1 ppm (mg/kg).

Fe, Cr, Co, Ni, Cu, Zn, Se, and Mo were analyzed by inductively coupled plasma mass spectrometry (ICP-MS, Varian Iberica; Madrid, Spain), which can analyze elements in a liquid matrix with sensitivity lower than 0.2 ppb (µg/kg). The samples were diluted 1:10 in a solution of 0.05% EDTA and 0.5% nitric acid to which 10 ppb of a solution mixture of elements used as internal standard was added (Sc, Y, Pt, Pd and Rh). The blanks and standards were prepared using the technique of additions on cow plasma also diluted 1:10 in the previously mentioned EDTA-nitric solution. In order to check the accuracy of the analytical method, a multielement standard solution (Merck, Temecula, CA, USA) with different concentrations (0, 10, 50 and 100 ppb) was used for the calibration. In all cases, the recovery rate was between 82.9% and 104.7%.

## 3. Animals and Legal Regulations

All four- to five-year-old Lidia bulls from different breeding farms used in this study fought in different Arena places under the regulation of the local legislation law [23], and at the moment of the sampling, all animals were already dead. In this way, we did not impose any additional experimental procedures that should cause any suffering or pain. All experimental procedures were performed in compliance with the provisions of the EU directive regulating the use of animals for scientific purposes [25] and the decree that regulates experimentation and animal protection in Spain [26].

## 4. Statistical Analysis

Data were tested for normality and analyzed using the SPSS 20.0 (SPSS Statistics for Windows, 20.0, IBM Corp, New York, NY, USA) statistical program. Descriptive analysis were performed indicating the mean value, standard deviation (SD) and minimum and maximum ranges.

## 5. Results

The mean values, standard deviation and maximum and minimum plasma concentrations of macrominerals in this study (Table 1) were greater than values for sodium reported in the literature, with a considerable variation between the minimum (87.33 mmol/L) and the maximum (248.68 mmol/L) concentrations. Potassium values also clearly increased (Table 1), although to a lesser degree, but the interquartile ranges were wide (Figure 1), while the calcium concentrations (2.96 mmol/L) were between the levels considered as normal, and the SD (0.31) and interquartile showed a closer range (Figure 1). The graphical representation of natremia is shown in Figure 2.

With regard to microminerals (Table 2), chromium showed increased mean values (1.24 mmol/L), whereas cobalt mean concentrations (0.041 mmol/L) decreased with regards to published dates. The graphical representation of some trace elements is shown in Figure 3 and Figure 4. Variations between the maximum and minimum concentrations of iron made the SD (13.67) and the interquartile ranges reach greater values of all the microminerals (Table 2 and Figure 3). Chromium, copper and nickel mean concentrations were greater than basal values reported in literature, and the SD of the last two trace elements (4.84 and 5.59) reflects the differences between the minimum and maximum values found (Table 2).

## 6. Discussion

### 6.1. Macrominerals

In the actual research, except for calcemia located in the usual range for cattle and for Lidia bulls, the plasma concentrations of other macrominerals analyzed (Mg, P, K and Na) after the fight were greater than those considered physiological for cattle [2,4,5]. When comparing our data with published studies obtained from animals of the same breed and in similar situations, most studies reported increased values [19,22,40]. The increases found in our study could be due to muscle cell destruction, the intense dehydration and hemoconcentration in the sampled bulls [27], in addition to the greater need to obtain energy during the intense exercise performed, it can consume significant amounts of minerals such as P and Mg, involved in obtaining ATP. The majority of the references consulted indicate an increase in phosphatemia and kalemia after exercise, regardless of the species sampled [8,19,29]. However, in some studies no changes in the concentration of Ca [8,29], Mg [30,48] and Na in blood were reported [48].

The lower levels of magnesemia in some bulls could be explained by the loss of this ion through sweat, as has been shown for horses after exercise, a situation that is observed particularly in these animals, with an adaptation mechanism known as short-term acclimatization of the sweat gland [8]. The increase in potassium in bulls after fighting could be justified by losses that occur in muscle fibers [49]. For this reason, during exercise, some muscle fibers intervene in muscular effort, while others remain inactive and are responsible for eliminating excess potassium. Increased muscle effort causes fiber recruitment, altering the elimination of excess ions and resulting to hyperkalemia [8]. However, in horses, when the exercise lasted more than 2 min, the plasma potassium concentration reached a plateau due to the presence of catecholamines and their effect on muscle fibers [8].

In Carpintero’s et al. opinion [27], the increase found in phosphatemia is due to the organic redistribution of phosphorus, as a consequence of respiratory acidosis and, on the other hand, of lactic acidosis caused by the intense exercise carried out by bulls. In addition, the concentration of K and Na in blood may increase, since in metabolic acidosis, reported in Lidia bulls because of the physical wear and tear produced by the fight [3], K and Na leave the intracellular spaces and are replaced by hydrogen ions (H^+^) [20].

Our results are in accordance with Alonso et al. [28] and Capen [50], who reported that the concentration of Ca^2+^ increased when serum pH decreased, because of competition by H^+^ ions to bind to negatively charged sites. In Lidia bulls, metabolic acidosis, which was previously mentioned, was already reported by Carpintero et al. [27] and confirmed by Escalera-Valente et al. [20], who found increased lactate concentrations following intense exercise. Our results also agree with those of Sánchez et al. [51], who also reported greater calcemia values in Lidia females after transport and stressful handling procedures. On the other hand, these authors and our study results do not agree with the decrease in plasma calcium and phosphorus described in horses as a consequence of greater excretion of minerals via the urine, the feces and the lower efficiency of mineral absorption in the intestine [8]. In horses, perhaps the most important reason is that, during exercise, the production of parathyroid hormone by the parathyroid glands decreases, which leads to a decrease in free calcium and phosphorus levels [8].

### 6.2. Microminerals or Trace Elements

There are not too many references with which to contrast the values found, and especially in Lidia cattle, since little research has been carried out, and exclusively on Se and Fe. Therefore, we are forced to compare the values found with those of cattle of different breeds. In addition, significant interferences occur between the minerals; therefore, the values of some depend on interactions between them (Mulder’s Wheel or Mulder’s Graph, cited by Malavolta) [52]. For example, in ruminants, Cu is the trace element with the greatest number of antagonists, the main one being Mo, followed by Fe, S and Zn [2,4,6].

The average of values of Fe and Cu are slightly greater than those reported for cattle, while those of Cr and Ni are much greater. On the contrary, the values of Zn, Se and Mo are similar and those of Co are much lower.

The levels of iron found in bulls after the fight (36.87 ± 13.67 μmol/L) is greater than those reported for diverse breeds [2,4,36] and similar to the maximum one (44.75 μmol/L) published by Mayland et al. [41] and Herdt and Hoff [38]. Only Bartolomé et al. [30] reported plasma concentrations of this mineral in the Lidia cattle, but their values were much lower than the values we found (17.71 μmol/L). Taking into account that more than half of the iron is a constituent of hemoglobin, which is the major protein in red blood cells [4,6], the increase in plasma iron after the bullfight would be due to the hemolysis consequence of the intense exercise performed. On the contrary, horses sampled after an international jumping competition showed a decrease in serum iron [8]. These authors suggested that the decrease could be due to greater urinary excretion, sweating and micro bleeds, as a consequence of the breakage of capillaries due to increased alveolar pressure and the relative slowness of the ionic absorption of the gastrointestinal tract.

Strenuous exercise, transport or infection can raise dietary needs for chromium in cattle, possibly as a result of increased losses of this mineral through urine [53]. Berger et al. [54] described an increase in chromium levels in athletes before and during the marathon test, verifying that the levels decreased in samplings made after the race had finished. However, the values found in this research (1.24 ± 0.58 μmol/L) are greater than those described by Subiyatno et al. [42] and Mayland et al. [41]. Anderson [55] reported that chromium, once mobilized, is quickly excreted in the urine, and although there could be multiple urination events during bullfighting, these could probably not be enough to eliminate the mobilized chromium. In the animals sampled, the physical exercise causes dehydration, and the kidney function tends to conserve water, concentrating the chromium. Additionally, in the case of the Lidia bulls, there would not be enough time to recover normal chromium values, since the fight ends with the death of the animal.

There are not many references on the blood concentrations of cobalt in cattle, and we have not found any in this breed, either under normal conditions, or after intense exercise. Thus, the values we found are lower than those described in milking bovine breeds [4,38,43,44]. Suttle [6] showed that cobalt has an important role on energy metabolism, and energy needs increase considerably during intense exercise such as during the fight, and this could lead to a decrease in plasma cobalt levels. Thus, cobalt participates in the formation of vitamin B12, an important growth factor for some ruminal microorganisms [7,56,57,58], increasing the production of NEFA and especially propionate. On the contrary, Berger et al. [54] found no differences in cobalt concentrations in human athletes before or after running a marathon. This may be because cobalt does not have the same importance for the production of energy in man, as a monogastric, as it does in ruminants.

Nickel values found in our study are greater than those described in cattle by Spears et al. [45] and Mayland et al. [41], who reported levels of 2.4 ppb, and a range of 1 to 6 ppb, respectively. Rumen bacteria inevitably required nickel for their survival; therefore, this microelement determines ruminal metabolism [6], while Stangl and Kirchgessner [59] comment that nickel is involved in the metabolism of lipids, particularly in the synthesis of phospholipids. We believe that the high levels of nickel found are due to the use of this trace element to maintain the increased metabolic activity produced by the physical exercise demands during the fight and, as previously mentioned, since the fight ends with the death of the animal, there is not enough time for the restoration of nickel blood levels. However, Berger et al. [54] have cited a decrease in blood nickel levels in human athletes after a race, although without statistical differences with the values found before starting the exercise.

The plasma selenium concentrations found are lower than those described by some research [4,38,39,46,47,60], but slightly greater than those described by Hesketh et al. [46] in other bovine breeds. This decrease in selenium levels in blood was also described in the Lidia breed [19,21,40,61]. However, the selenium values found in the present study are still lower than those described by García-Belenguer et al. [40] in Lidia cows. However, our data are greater than those recorded by García-Belenguer et al. [21], probably due to the greater interest and care that exists in recent years to supplement the diet of animals reared in extensive farming systems, adding mineral and vitamin correctors to their diet, and especially selenium.

García-Belenguer et al. [21] reported that selenium levels may be diminished as a significant amount of this trace element is used for the synthesis of GSH-Px, an enzyme that develops a protective function against oxidative damage [7,62,63]. Another possible explanation for the decrease in blood values of this trace element is that selenium is contained in macrophages [64], and these increase considerably with stress produced in handling, transport and the fight itself [51,61,65]. This hypothesis is based on the fact that we have found a statistically significant correlation between selenium and cortisol concentrations [66].

The value for cupremia found in this study is slightly greater than that cited [4,38,51] and much greater than levels reported by Kaneko et al. [2] for bovines. However, it is close to the value indicated by Mayland et al. [41] and Hesketh et al. [46]. The increase in cupremia values may be because copper participates in erythropoiesis and the protection of tissues from oxidative damage, and these functions markedly increase as a consequence of the exercise performed by the animals during the fight. The physical effort causes a greater need for tissues oxygenation, which in turn induces the production and, above all, the splenic release of erythrocytes. In addition, both aerobic and anaerobic metabolism increases, producing a large amount of hydrogen ions that cause oxidative damage [20]. García-Belenguer [40] stated that this breed has a greater cupremia level than considered normal in cattle, and since stress and intense exercise can produce transient hypercupremia, this could explain the abnormally increased values we found.

Zn-containing enzymes are found in all of the major pathways involved in carbohydrate, lipid, protein, and nucleic acid metabolism [7]. Several authors point out that plasma zinc contents are particularly susceptible to stress [67]. Nevertheless, this does not agree with the findings in this research, since zinc levels in the bulls are greater than those cited for bovines [41,44]. Greater values have been indicated by other researchers in dairy cows affected by inflammation of mammary gland [68] or in much younger cattle (3 to 5 months old) experimentally infected with bovine spongiform encephalopathy, although differences in this study were not significant between control and sick animals [46].

In the opinion of Cousings [69], most of the zinc circulating in the bloodstream is located inside the red blood cells (up to 80%), and taking into account the exercise hemolysis described by several authors [70,71,72], which could occur in the bull during the fight, would explain the increased zinc plasma concentration found in our research.

The concentrations of molybdenum found (0.11 ± 0.08 μmol/L) are below those considered normal in bovine and ovine species [4,46]. In human athletes, Berger et al. [54] did not find significant differences in molybdenum concentrations before and after a marathon. Perhaps the decrease in molybdenum levels is determined by the increase in copper, which, as we have indicated, is slightly greater. If Lidia cattle have high copper values under normal conditions, and exercise and stress situations can cause transient hypercupremia, this could explain the abnormally low molybdenum concentration that agrees with the results of García-Belenguer [40] and Agüera Buendía et al. [19]. We must not forget that the cattle needs of molybdenum are extremely low and that copper is a molybdenum antagonist [6,7].

In conclusion, our study showed the effect of bullfighting (intense physical exercise) on the Lidia breed, resulting an increased macrominerals. In addition, iron, chromium, copper and nickel increased, compared to values considered “normal” for cattle, while cobalt decreased, and zinc, selenium or molybdenum were similar. Although other causes have been cited, in our opinion the most important reason for the increase of these minerals could be related to the dehydration and subsequent hemoconcentration in the bulls as a consequence of the intense exercise of the bulls.

## Figures and Tables

**Figure 1 vetsci-08-00097-f001:**
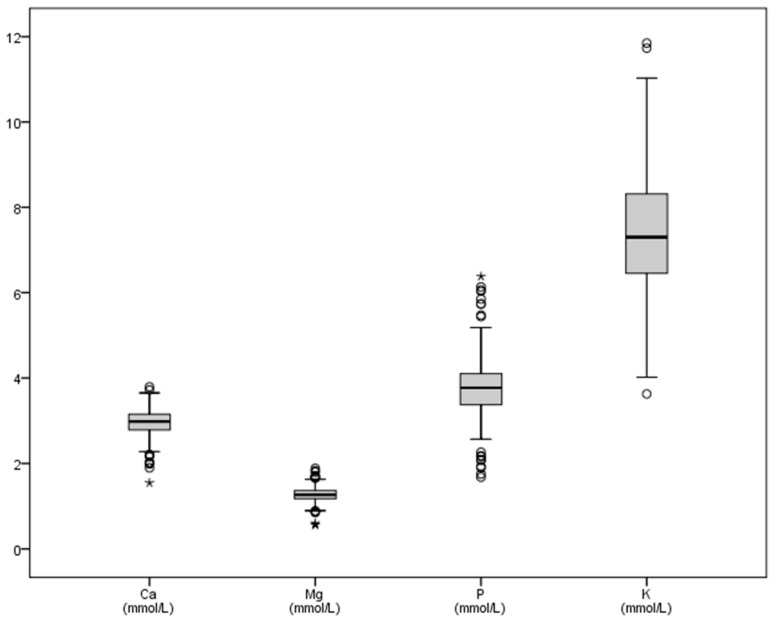
Graphic representation of some macrominerals (calcium, phosphorus, magnesium, potassium) in bull’s plasma after physical exercise. Ca: calcium, P: phosphorus, Mg: magnesium, K: potassium. All variables in mmol/L. Some outliers are not shown in the graph.

**Figure 2 vetsci-08-00097-f002:**
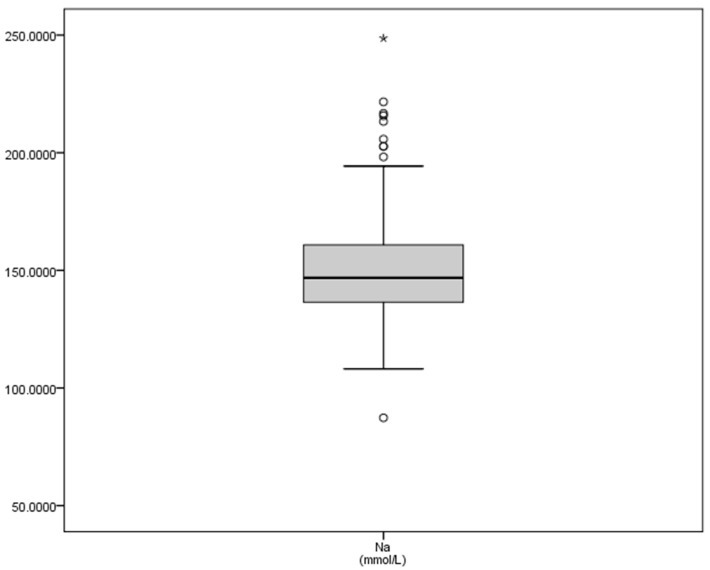
Graphic representation of the natremia of bulls after physical exercise. Footnotes: Na: sodium, in mmol/L.

**Figure 3 vetsci-08-00097-f003:**
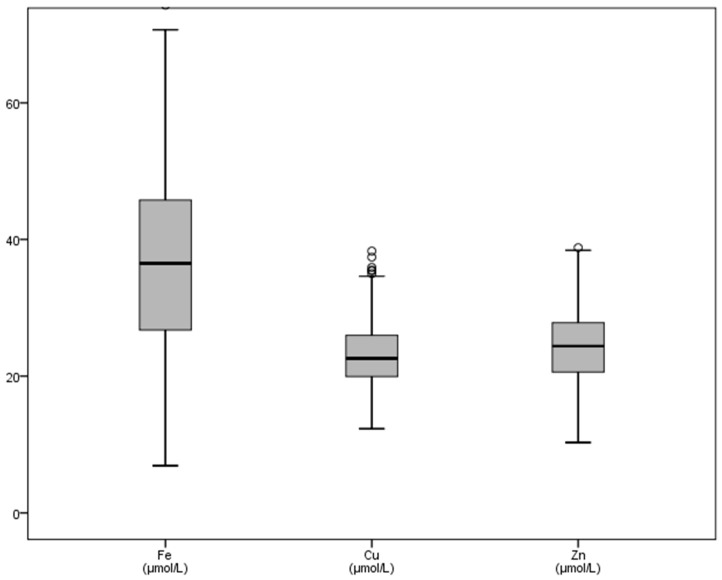
Graphic representation of some trace elements (iron, copper, zinc) in bull’s plasma after physical exercise. Footnotes: Fe: iron, Cu: copper, Zn: zinc. All variables in μmol/L. Some outliers are not shown in the graph.

**Figure 4 vetsci-08-00097-f004:**
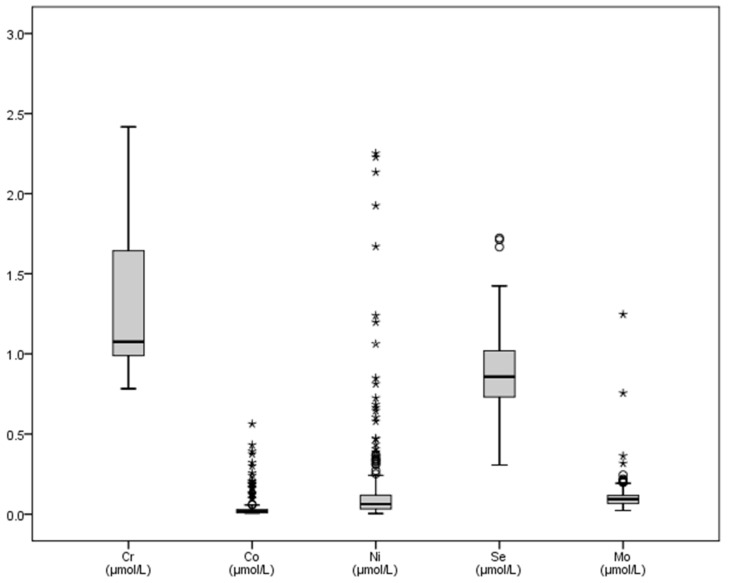
Graphic representation of some trace elements (chromium, cobalt, nickel, selenium, molybdenum) in bull’s plasma after physical exercise. Footnotes: Cr: chromium, Co: cobalt, Ni: nickel, Se: selenium, Mo: molybdenum. All variables in μmol/L. Some outliers are not shown in the graph.

**Table 1 vetsci-08-00097-t001:** Mineral values (macrominerals) in bull plasma after intense exercise: descriptive statistics. Values indicated by other authors are also shown.

	Ca	Mg	P	K	Na
N	438	438	438	437	437
Mean	2.96	1.27	3.78	7.50	150.15
SD	0.31	0.17	0.65	1.58	19.59
Minimum	1.55	0.56	1.68	3.63	87.33
Maximum	3.79	1.88	6.38	18.30	248.68
Sum	1296.85	557.62	1656.22	3279.71	65,616.02
Range	2.25	1.33	4.70	14.67	161.35
Some authors ^a^	2.00–2.43	0.70–1.23	1.45–2.03	3.3–5.8	132–155
Radostits et al., 2006 [4]	2.43–3.10	0.74–1.10	1.08–2.76	3.9–5.8	132–152
Kaneko et al., 2008 [2]	2.43–3.10	0.74–0.95	1.81–2.10	3.9–5.8	
Carpintero et al., 1996 [27] ^b^	2.80–2.83	1.15–1.31	3.85–3.86	—	
Alonso et al., 1997 [28] ^b^	3.15	1.19	4.01	8.10	162.8
Chaves et al., 2001 [29] ^b^	2.61	—	—	9.49	160.1
Bartolomé et al., 2005 [30] ^b^	2.87	1.07	3.18	—	141.8

Ca (calcium), P (phosphorus), Mg (magnesium), K (potassium), Na (sodium), SD (standard deviation). All variables in mmol/L. ^a^ Some authors, in various autochthonous Spanish breeds [31,32,33,34,35]. ^b^ Everything in Lidia cattle.

**Table 2 vetsci-08-00097-t002:** Mineral values (trace elements) in bull plasma after intense exercise: descriptive statistics. Values indicated by other authors are also showed.

	Fe	Cr	Co	Ni	Cu	Zn	Se	Mo
N	327	427	255	427	427	427	427	427
Mean	36.87	1.24	0.041	0.249	22.63	24.14	0.886	0.111
SD	13.67	0.58	0.07	1.07	4.84	5.59	0.210	0.08
Minimum	6.90	0.78	0.007	0.004	11.10	10.30	0.307	0.023
Maximum	80.10	9.92	0.56	13.86	41.80	39.70	1.723	1.25
Sum	12057.70	529.96	10.48	106.11	9662.8	10307.5	378.39	47.34
Range	73.20	9.13	0.56	13.85	30.70	29.40	1.42	1.22
Some authors ^a^	23.2–44.7	0.01–0.06	3.32–4.24	0.04–0.102	12.5–25.9	12.2–46.1	0.55–1.4	0.99 ± 0.03
Radostits et al., 2006 [4]	10–29	—	0.17–0.51	—	15.7	21.35	—	0.52
Kaneko et al., 2008 [2]	10.2–29.0	—	—	—	5.16–5.54	—	—	—
Underwood and Suttle, 2002 [36]	17.4 ±5.2	—	—	—	3–9	—	—	—
Araujo, 2008 [37]	23.2–44.7	—	—	—	—	12.2–38.2	2.6–15.2	—
Herdt and Hoff, 2011 [38]	19.7–44.7	—	0.029–0.34	—	9.42–17.27	9.18–290.7	0.823–1.772	0.02–0.37
Rollin and Guyot, 2013 [39]	—	—	—	—	13–18	14–21	1.02–1.78	—
García-Belenger et al., 1991 [40] ^b^	—	—	—	—	—	—	0.13–0.48	—
Bartolomé et al., 2005 [30] ^b^	17.71	—	—	—	—	—	—	—

Fe (iron), Cr (chromium), Co (cobalt), Ni (nickel), Cu (copper), Zn (zinc), Se (selenium), Mo (molybdenum), SD (standard deviation). All variables in μmol/L. ^a^ In different breeds. (Fe: [41], Cr: [41,42], Co: [43,44], Ni: [41,45], Cu: [41,46], Zn: [41,44,46], Se: [46,47], Mo: [46]. ^b^ In Lidia cattle.

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
