# Peer review of "Effect of Intense Exercise on Plasma Macrominerals and Trace Elements in Lidia Bulls"

_vetsci, 2021, doi:10.3390/vetsci8060097_

Round 1

Reviewer 1 Report

COMMENTS TO AUTHRORS

Revision of the Article entitled “Effect of Intense Exercise on Plasma Macrominerals and Trace Elements in Lidia Bulls”.

The Authors measured plasma concentration of macro and microminerals in 438 male Lidia bulls after intense physical exercise during a bullfight. Then, they compared their results with reference value reported in literature for cattle. They assessed that macrominerals and Cr, Ni, Cu and Zn are greater than published data while Co, Se and Mo are lower than values reported for other bovine breeds. They stated that these changes could be due to muscle cell metabolism, hepatic need to provide energy, and intense dehydration and haemoconcentration by losses through sweat glands or urination.

ABBREVIATION

Line 33: please add ICP-MF

INTRODUCTION

Line 43: please change mg with Mg

Line 41-49: please add which are macrominerals and which are microminerals

Line 50-56: please add references regarding cattle or write that these statement are reported for horses

Line 58: please modify the references for intense physical exercise: reference 9 regards show jumping horses and their work is not consider “intense”; Reference 10 investigated the potential bioaccumulation of some minerals in horses from the industrial risk area of Sicily not the modification due to exercise.

Line 60: please specify studies that measured macrominerals in the Lidia breed after exercise or stressful situations.

MATERIALS AND METHODS

Line 71-72: just a curiosity: are all the bulls slaughtered? Also the winners? Or only the losers due to the lesions reported?

Line 98: please specify which statistical analysis were performed

RESULTS

Please reformulate it more clearly. Please specify the literature to which you compare your results and report the data of the literature. Did you only make a subjective comparison of the means or did you perform a statistical analysis to asses that your values are higher/lower than the values reported in the literature? If you did a statistical analysis, please report the p-value.

Line 103: are all data normally distributed?

Line 103-106: did you refer to all macrominerals or only to sodium?

Tables: specify what “Valid N” is

Figures: they do not add any information to the text and tables

DISCUSSION

In the introduction you wrote “The aim of the present study was to obtain reference mineral blood concentrations in bulls after intense exercise”, however you did not give reference ranges. Moreover, you wrote “Knowledge about mineral changes could improve our under standing of the Lidia bull, and perhaps extend to other animals in the wild that are difficult to handle during stressful situations and thus better manage their welfare”, however you did not discuss about this.

If you did only a subjective comparison without any statistical analysis, I do not agree that you can state that your results are greater/lower than others. Moreover, I do not think that is correctly to compare your results with the literature, in particular with data reported for other breeds. Are you sure that the breed and sex do not influence the values? I understand that it is difficult to obtained sample from live bulls before the fight to do a direct comparison before and after exercise on the same sample. However, if there are not reference values for Lidia bulls, maybe it was better to first assess basal values in Lidia male bulls and then evaluate minerals after exercise in order to can make a right comparison (For example how it was done in the study “reference 34”). Otherwise, you must take into account that are data reported for different breeds and discuss it and, in my opinion, you can not state that the value are greater/lower due to exercise but only that probably the value are different.

Please specify always if published data regard Lidia bulls or other breeds and report the published values.

Line 122 and 147-152: here you reported that your data are in accordance with the literature that had found an increased calcaemia but in the results section you reported that your value of calcium concentration was between normal ranges.

Line 138-140: reference 7 regards horses not Lidia bulls. Moreover, did you find a decreased or increased in magnesaemia? If you found hypermagnesaemia, please discuss the difference with the cited literature.

Author Response

Dear Editor, Dear Referees,

We greatly appreciate the reviewers for their work and dedication. This will undoubtedly lead to a higher quality manuscript, making it easier to read and understand.

We will try to respond to each of the reviewers' indications, as well as introduce the appropriate changes in the final version of the manuscript (in blue colour).

Thank you.

Reviewer 2 Report

Dear authors,

I read your work with great interest and attention. However, I propose these changes to the present version, in order to facilitate the reading of the work and to increase the effectiveness of its scientific message.

Abstract

I coud not agree with the idea underlinied in yellow.

Line 17 - Minerals are an important part of animal metabolism and are either directly or indirectly involved in a number of biological functions.

I suggest this new redaction,...

Minerals are inorganic substances, present in all body tissues and fluids and their presence directly or indirectly influences the maintenance of certain metabolical processes which are essential to the development of several biological functions.

Lines 24-27

All macrominerals, (Ca: 2.96 24 ± 0.31, Mg: 1.27 ± 0.17, P: 3.78 ±0.65, K: 7.50 ± 1.58, Na: 150.15 ± 19.59 in mmol/L), some microminerals (Cr: 1.24 ± 0.58, Cu: 22.63 ± 4.84, Zn: 24.14 ± 5.59, in μmol/L) and ultratrace minerals (Ni: 0.249 ± 1.07, in μmol/L) showed greater mean values than reported reference values in the published literature. Other trace minerals  values as Co: 0.041 ± 0.07, Se: 0.886 ± 0.21 and Mo:0.111 ± 0.08, in μmol/L were lower than those reported for other bovine breeds.

Point 1 - It is necessary to justify the high values and the reduced values.

Point 2 - Are the values compared with the reference values of this or other breeds?

Introduction

Line 45 - homeostasis instead state????

Could alter the animals homeostasis 

Point 3 - The role of minerals is briefly covered in the introduction. In my opinion, it is justified to emphasize the role of minerals in the metabolism of ruminants and animals subject to very demanding efforts in a short period of time, whether they are human athletes, horses or dogs used in races. In this sense, it is also important to mention the consequences of their imbalances, about which the text is silent.

Point 4 - A paragraph on the comparison of the mineral values previously described in this (Herrera et al,  1990, and Carpintero et al. 1996, had previously studied some macrominerals in this breed) and other  breeds of ox should be written.

Materials and Methods

Could you please describe the blood colection method?

Are all animals transported under the same conditions  from their farms to the arena?

How long does it last between the last meal, the run and between the last meal and the slaughter?

Was the time between the animal's entry into the arena and the slaughter the same?

Was the duration of combat equal for all animals? What about the time between the end of the race and the blood sample collection?

Results

The description of the results seems confusing and needs to be improved.

Why don't your tables compare the known values of minerals before and after exercise in this species and preferably in this breed?

Discussion

In my opinion you has a very  good scientific discurs but you need to reorganize the logical sequence of your ideas in the text. 

Please uniformise your terminology

Line 126 - and K (kalemia)

Line 144 - resulting in hyperkalaemia

Line 128 - Mg (magnesemia) 

Line 138 - magnesaemia 

Line 128 - changes in Ca (in calcemia)

Line 152 - calcaemia values in Lidia females

Lines 125-128

The majority of the references consulted indicate an increase in P (hiperphosphatemia) and K (hiperkalemia) after exercise regardless of the species sampled [7,8,20]. However, in some studies no changes in Ca (hiper calcemia) [7,29], Mg (hipermagnesemia) [30,31] and Na (hipernatremia) [31] are reported.

Because P means Phosphorus, K means  Potassium, etc....

Lines 133- 137

Reorganize the text, here you are my suggestion,...

In metabolic acidosis reported in Lidia bulls due to physical wear produced by the fight  [21], potassium  and sodium leaves the intracellular spaces and are replaced by hydrogen ions (H+) [2].

At this point please when you are describing potassium or any other mineral write everything about them in sequence,....in the same paragraph,....

Lines 152-156

On the other hand, these authors and our study results do not agree with the decrease in plasma calcium and phosphorus levels  described in horses as a consequence of greater excretion of minerals via the urine, the faeces and lower efficiency of mineral absorption in the intestine [7]. 

Lines 160-161

Significant interferences occur between minerals, the values of some depending on interactions between them (Mulder's wheel or Mulder's Chart, cited by Malavolta).

Conclusions

Please explain the reasons for decreased mineral levels.

Tables

Consider compare your values with reference values of the specie and compare your values with other similar works.

Good work

Author Response

Dear Editor, Dear Referees,

We greatly appreciate the reviewers for their work and dedication. This will undoubtedly lead to a higher quality manuscript, making it easier to read and understand.

We will try to respond to each of the reviewers' indications, as well as introduce the appropriate changes in the final version of the manuscript (in blue).

Thank you.

Reviewer 3 Report

Congratulations to the authors for this article that is an important contribution for Lidia bulls. Some questions may be answered in the manuscript. 

1) In material and methods:

IN HOW MANY SEASONS HAVE THE SAMPLES BEEN TAKEN?

2)DID THE AUTHORS DO ANY ADDITIONAL HEMATOLOGICAL STUDIES?

3)Bibliographic reference number 17 is underlined in yellow by mistake.

Author Response

(The authors gave the same response as above.)

Round 2

Reviewer 1 Report

Dear authors,

the revised version of the manuscript has greatly improved. I only suggest some more little changes.

Line 67-69 I suggest to add also that for macrominerals there are few data.

Line 70 Please change “obtain reference” with “evaluate”. If you write "reference", I expect that you statistically calculate the “reference intervals”.

Line 164-165 Please change “1” and “2” with “a” and “b” in the footnotes of the table.

Line 224-229 There is a repetion of a paragraph

Line 231 I suggest to change with “There are very few…”. It is better and stronger than “There are not much…”

Line 257 Please add “in human athletes”

Author Response

Dear Referee 1.

Thank you for your cooperation. We have made the suggested modifications.

Line 67-69 I suggest to add also that for macrominerals there are few data.

Done. 

Line 70 Please change “obtain reference” with “evaluate”. If you write "reference", I expect that you statistically calculate the “reference intervals”.

Done.

Line 164-165 Please change “1” and “2” with “a” and “b” in the footnotes of the table.

Done.

Line 224-229 There is a repetion of a paragraph.

Sorry, but we could not locate the repeated paragraph, possibly because there has been some change in the numbering of the lines. I beg the editor to remove it if he finds it.

Line 231 I suggest to change with “There are very few…”. It is better and stronger than “There are not much…”

Done

Line 257 Please add “in human athletes”

Done